# Effects of Organic Biostimulants Added with Zeolite on Zucchini Squash Plants Infected by Tomato Leaf Curl New Delhi Virus

**DOI:** 10.3390/v14030607

**Published:** 2022-03-15

**Authors:** Livia Donati, Sabrina Bertin, Andrea Gentili, Marta Luigi, Anna Taglienti, Ariana Manglli, Antonio Tiberini, Elisa Brasili, Fabio Sciubba, Gabriella Pasqua, Luca Ferretti

**Affiliations:** 1Council for Agricultural Research and Economics–Research Centre for Plant Protection and Certification, 00156 Rome, Italy; sabrina.bertin@crea.gov.it (S.B.); andrea.gentili@crea.gov.it (A.G.); marta.luigi@crea.gov.it (M.L.); anna.taglienti@crea.gov.it (A.T.); ariana.manglli@crea.gov.it (A.M.); antonio.tiberini@crea.gov.it (A.T.); luca.ferretti@crea.gov.it (L.F.); 2Department of Environmental Biology, Sapienza University of Rome, 00185 Rome, Italy; elisa.brasili@uniroma1.it (E.B.); fabio.sciubba@uniroma1.it (F.S.); gabriella.pasqua@uniroma1.it (G.P.); 3NMR-based Metabolomics Laboratory (NMLab), Sapienza University of Rome, 00185 Rome, Italy

**Keywords:** Begomovirus, ToLCNDV, pest management, plant fitness, metabolic response, antiphytoviral activity, *Cucurbita pepo* L.

## Abstract

The use of organic substances in integrated pest management can contribute to human- and environment-safe crop production. In the present work, a combination of organic biostimulants (Fullcrhum Alert and BioVeg 500) and an inorganic corroborant (Clinogold, zeolite) was tested for the effects on the plant response to the quarantine pest tomato leaf curl New Delhi virus (ToLCNDV). Biostimulants were applied to healthy and infected greenhouse-grown zucchini plants, and the vegetative parameters and viral titer were evaluated. Although no antiviral effects were observed in terms of both virus replication and symptom expression, these biostimulants were shown to influence plant fitness. A significant increase in biomass and in leaf, flower, and fruit production was induced in both healthy and infected plants. Biostimulants also enhanced the production of metabolites commonly involved in plant response to virus infection, such as carbohydrates, phenylpropanoids and free amino acids. These results encourage new field trials to evaluate the actual productivity of infected plants after treatments and the possible application of organic biostimulants in agriculture.

## 1. Introduction

The growth in the world population constantly challenges the agricultural sector to increase crop yields and the efficiency of resource exploitation. Fertilizers and pesticides are still widely employed to achieve the required production standards, although their active ingredients are often persistent pollutants in soil and wastewater and are toxic to various organisms. Over the past three decades, several technological innovations as well as eco-friendly alternatives to agrochemicals have been proposed to improve the sustainability of agriculture and reduce its impact on both ecosystems and human health [1]. Natural plant biostimulants are generating considerable interest as new-generation products that combine the actions of plant growth promotors and stress alleviators [2]. Practically, these products contribute to the flowering, growth, fruit set, productivity, and nutrient use efficiency of crops as well as to the tolerance against a wide range of abiotic stresses [3].

Among the available biostimulants, Fullcrhum Alert, which is a liquid extract of alfalfa (*Medicago sativa* L.), brown algae and molasses, has been commercialized as a bio-inducer acting as an elicitor. The extract of alfalfa is rich in flavonoids and saponins, which are already known to be active against a wide range of crop pests [4]. Seaweed extracts, especially brown algae, are commonly present in biostimulant formulations because they contain bioactive compounds such as phytohormones, microelements, algal-specific polysaccharides, betaines, polyamines, and phenolic compounds [5]. Another commercially available biostimulant is Fullcrhum BioVeg 500, an organic compound containing glycine betaine, known for playing an important role in balancing metabolic dysfunctions caused by countless stressful conditions such as stress salinity, heavy metal contamination, and drought [6,7,8,9,10]. These biostimulants are often used in combination with zeolite, a soil conditioner that enhances soil water retention, the release of important macronutrients such as phosphate and calcium, and the absorption of toxins and heavy metals [11]. Although the role of biostimulants to enhance plant fitness is well-known, the available information on their effect on systemic infection by plant pathogens is limited. While other compounds are already known to induce protection against fungi, bacteria, and viruses [12], the biostimulants are not likely to play a direct role against pathogens, but they can trigger plant defense priming [13].

Cucurbits are globally cultivated and represent economically important crops [14]. Several virus diseases have been reported to infect cucurbits, with a negative impact on greenhouse and open field production. The recent emergence of tomato leaf curl New Delhi virus (ToLCNDV) (genus Begomovirus, family Geminiviridae) in the Mediterranean region has caused major problems in terms of crop yields and quality. The virus is transmitted by the whitefly *Bemisia tabaci* Gennadius (Hemiptera: Aleurodidae) in a persistent manner, and until now it has been reported in Italy, Spain, France, Portugal, Greece, Tunisia and Algeria [15,16,17,18,19]. It has been categorized as a quarantine pest (EU regulation 2019/2072, Annex IIB) and included in the EPPO Alert List. ToLCNDV isolates from Spain and Italy were characterized and shown to share a high degree of homology among themselves and were considered as a unique strain, named ToLCNDV-ES [20,21]. This strain seems to prefer cucurbit rather than solanaceous hosts, and extensively infects zucchini squashes. Infected plants show yellow mosaic, leaf curling, vein swelling and plant stunting, and fruit skin may present a rough appearance with longitudinal cracking [15,22].

This study aimed at assessing the effects of organic biostimulants, in combination with zeolite as a soil conditioner, on zucchini squash plants infected by ToLCNDV, for possible application in sustainable programs of disease prevention and protection. In this context, we evaluated possible effects on viral titer as well as on plant fitness, estimated as leaf, flower, and fruit production and biomass weight. Since the mode of action of biostimulants is still largely unknown, metabolic changes associated with treatments were also investigated.

## 2. Materials and Methods

Seeds of *Cucurbita pepo* var. ‘Tiziano’ were sown into 10 cm plastic pots containing fresh soil Completo^®^ (Vigorplant). The plants were grown under controlled conditions of temperature (24 ± 1 °C), photoperiod (16/8 h light/dark), and relative humidity (55%) with watering as required. Care was taken to ensure that the plant size was as uniform as possible. A pool of the seed batch used in the experiments was tested by qPCR [23] to exclude any ToLCNDV infection or contamination.

The source of ToLCNDV-ES inoculum was a symptomatic zucchini squash plant collected in an open field in Terracina (Latium, Italy) during the summer of 2019 (isolate CREA-Terr-2019). The zucchini plant was tested for the most common cucurbit potyviruses (zucchini yellow mosaic virus, watermelon mosaic virus, cucumber mosaic virus) to exclude any mixed infections. The isolate was maintained through several mechanical inoculations on healthy zucchini squashes, as follows. Young and highly symptomatic leaves were ground in phosphate buffer (1:5, *w*/*v*) in extraction bags (Bioreba AG, Basel, Switzerland), and 80 µL of sap were applied by rubbing the cotyledons and first true leaves of healthy plants previously sprinkled with tricalcium aluminate. The leaves were then gently rinsed.

### 2.1. Experimental Design

Three independent experiment repeats with Fullcrhum (BioHelp Your Planet srl, Viterbo, Italy) biostimulants were performed. Each repeat consisted of a total of 60 zucchini plants split into four experimental theses: 15 healthy untreated (HU), 15 infected untreated (IU), 15 healthy treated (HT) and 15 infected treated (IT) plants. The plants of each thesis were kept separately in different screenhouses. Four biostimulant treatments were carried out per thesis, as indicated by the manufacturer. The first and second treatments were performed at the plant cotyledonary stage (T1) and one week later (T2), respectively, at both root and leaf level. The third (T3) and fourth (T4) applications were distributed every 15 days on leaves only. The radical treatment was carried out by distributing 40 mL/plant of a mixture containing Fullcrhum Alert/Fullcrhum BioVeg 500/Clinogold 1/1/1 (*v*/*v*/*w*) directly onto the soil. The leaves were nebulized with Fullcrhum Alert/Fullcrhum BioVeg 500/Clinogold 3/4/2,5 (*v*/*v*/*w*), by means of a 1 L manual vaporizer, taking care to wet the leaves uniformly. The chemical compositions of the used products were as follows: Fullcrhum Alert: 1% organic nitrogen, 10% organic carbon, 6% potassium oxide and 1% betaine; Fullcrhum BioVeg 500: >30% glycine betaine, 5% organic nitrogen and 15% organic carbon; Clinogold: 95% clinoptililote. The ToLCNDV inoculation on IU and IT plants was carried out between the first two treatments T1 and T2, as illustrated in Figure 1.

### 2.2. Vegetative Parameters and Symptom Evaluation

The number of leaves, flowers, and fruits as well as the fruit fresh weight (FW) and the plant dry weight (DW) were recorded 1 week after the fourth treatment for each thesis. The DW was evaluated by keeping the plants at 60 °C for 48 h. The appearance and evolution of ToLCNDV systemic symptoms on both IU and IT plants were observed at the end of the experiments, and a classification of symptom expression was used (Table 1). A score was assigned to each plant of the two infected theses.

### 2.3. Quantification of Virus Titer

The ToLCNDV titer was assessed by means of quantitative real time-PCR (qPCR) in ToLCNDV-infected plants (IU and IT theses) from the second until the fourth treatment (T2, T3 and T4). Healthy untreated (HU) plants served as negative control. At each time point, a disk from each last true leaf was collected from three plants of each thesis (IU, IT and HU) and pooled together. Total DNA (TDNA) was extracted using a CTAB-based extraction protocol [24]. TDNA was quantified in an ND-2000c Spectrophotometer (NanoDrop Technologies, Thermo Scientific, Wilmington, DE, USA) and diluted to a final concentration of 10 ng/µL. The qPCR standard curve was constructed on a cloned DNA-A fragment of about 1500 bp of the ToLCNDV-ES isolate used for the experimental inoculations. The fragment was amplified by end-point PCR using the specifically designed primer pair IntergA-FW (ATATGCATCGTTCGCCGTTTG) and IntergA-RV (TCTGTTCATGGGCCTGTTCG) and cloned using pGEM^®^-T Easy Vector (Promega Corporation, Madison, WI, USA). Ten-fold serial dilutions from 10^−2^ to 10^−6^ of the purified plasmids were analyzed in qPCR, and a calibration curve with *R*^2^ = 0.980 was obtained. The qPCR was performed in a C1000 Touch Thermal Cycler equipped with a CFX96 Real-Time System (Bio-Rad) using the primers ToLA-Up, ToLA-Low and TaqMan probe ToLA-Probe described by Simòn et al. (2018) [25] for the amplification of a 109-bp fragment of the AV2 gene in the DNA-A. Two µL of TDNA and of standard samples were amplified in a 20 µL mixture containing 2× TaqMan Universal PCR Master Mix (Applied Biosystems, Waltham, MA, USA), 500 nM of each primer and 50 nM of probe. The cycling conditions consisted of incubation at 95 °C for 10 min and 40 cycles at 95 °C for 15 s and at 60 °C for 1 min. Each sample was analyzed in triplicate, and the DNA extracted from HU plants was used as a negative control. The absolute quantification of the viral titer was calculated, based on the Cq of the samples, using the calibration curve [26]. The viral titer was quantified as ToLCNDV copies in 2 µL of reaction.

### 2.4. H-NMR Based Metabolomics of Zucchini Leaves

Zucchini leaves from the four experimental theses were collected at T2, T3 and T4, ground and immediately stored at −80 °C. A total of 0.5 mg from three ground leaves was extracted following a modified Bligh–Dyer protocol [27]. Briefly, each aliquot was placed in a mortar, ground in liquid nitrogen, and added to a cold mixture composed of chloroform, methanol, and water in a 2:2:1.2 (*v*/*v*/*v*) proportion. After overnight incubation at 4 °C, the samples were centrifuged for 25 min at 4 °C with a rotation speed of 11,000 g. The upper hydrosoluble phase and the lower hydrophobic phase were carefully separated and dried under a gentle flow of nitrogen. The hydrophilic phase was resuspended in a mixture of D_2_O/MeOD in a ratio of 2:1 (*v*/*v*) containing 3-(trimethylsilyl)-propionic-2,2,3,3-d4 acid sodium salt (TSP, 2 mM) as an internal chemical shift and concentration standard. The hydrophobic phase was resuspended in CDC^l3^ with hexamethyldisiloxane (HMDS, 2 mM) as an internal standard. All solvents and standards were purchased from Sigma Aldrich (St. Louis, MO, USA). All spectra were recorded at 298 K on a Bruker AVANCE III spectrometer operating at the proton frequency of 400.12 MHz and equipped with a Bruker multinuclear z-gradient inverse probe head. Hydroalcoholic ^1^H spectra were acquired employing the preset pulse sequence for solvent suppression with 128 transients, a spectral width of 6000 Hz and 64K data points for an acquisition time of 5.5 s. The recycle delay was set to 9.5 s to achieve complete resonance relaxation between successive scansions. ^1^H-NMR spectra of chloroform extracts were acquired employing a single pulse sequence with 128 transients, a spectral width of 6000 Hz and 32K data points for an acquisition time of 2.75 s. The recycle delay was set to 10.25 s to achieve complete resonance relaxation between successive scansions. Resonance assignment was carried out based on 2D ^1^H-^1^H TOCSY and ^1^H-^13^C HSQC experiments as described in Giampaoli et al. (2021) [28]. Only the identified molecules were considered for the study, and their quantification was performed by integration of their NMR signals. Due to the overcrowding of ^1^H NMR spectra, only those signals that did not overlap with other resonances were considered for integration. Quantities were expressed in µmol/g through comparison of the relative integrals with the reference concentration and normalized to the number of protons (TSP: 9 protons and HMDS: 18 protons) and to the fresh weight of leaves.

### 2.5. Statistical Analysis

The analysis of variance (ANOVA) on the four groups of plants (HU, IU, HT, IT) from the three independent experimental repeats was performed for each vegetative parameter: number of leaves, flowers, fruits and DW. Significantly different groups were identified using Fisher’s least significant difference (LSD) post-hoc test. The viral titer in IU and IT plants, as well as the fruit FW of IT and HT plants, were compared by Student’s *t*-test. Fold change (IT vs. IU and HT vs. HU) of metabolites was measured on the data matrix with the Unscrambler ver. 10.5 software (Camo Software AS, Oslo, Norway). Metabolites with fold changes above or equal to 1.5 were considered as significant.

## 3. Results

### 3.1. Effects of Biostimulants on the Vegetative Parameters

Several vegetative parameters of the plants belonging to the HU, IU, HT, and IT theses were recorded at the end of each experiment. The number of leaves recorded was significantly higher (*p* < 0.0001) in both treated healthy (HT: 14 ± 1.78) and ToLCNDV-infected (IT: 17 ± 1.48) plants than in the respective untreated theses (HU: 13 ± 1.94; IU: 11 ± 0.76). The difference in leaf number was particularly evident in case of virus infection, and the IT plants showed the highest values across the four theses (Figure 2A). 

The effects of treatments were observed also on the mean number of flowers (*p* < 0.05) and fruits (*p* < 0.01). In the absence of biostimulants, a significant reduction in the number of flowers was observed in the infected plants (IU: 11 ± 0.64) compared to the healthy ones (HU: 13 ± 0.08); after the treatments, the mean number of flowers significantly increased in the infected plants (IT: 14 ± 1.78) and was comparable to the healthy ones (HT: 14 ± 0.35) (Figure 2B). Both healthy and ToLCNDV-infected treated plants brought to maturity a mean number of fruits, which was significantly higher than those of the untreated theses (HT: 3 ± 0.35, IT: 3 ± 0.36, HU: 2 ± 0.02, IU: 1 ± 0.76). The difference in fruit number was particularly evident in the case of virus infection, since the IU and IT plants had the lowest and highest number of fruits across the four theses, respectively (Figure 2C). The variability in fruit production was associated with different degrees of fruit ripeness. The fruits of treated healthy and ToLCNDV-infected plants had an average fresh weight of 6.5 ± 0.28 and 4 ± 0.35 g, respectively (*p* = 0.05), while for both the untreated (HU and IU) theses, FW could not be assessed because fruits were unripened and poorly developed (Figure 3). Finally, Fullcrhum treatments also resulted in a significant increase in dry biomass (*p* < 0.0001). The healthy treated plants had the highest DW mean value (HT: 2.7 g ± 0.22), and the infected treated plants showed a DW (IT: 2.1 g ± 0.18) that was significantly higher than that of the infected untreated plants (IU: 0.5 g ± 0.15) and comparable to that of the healthy untreated plants (HU: 2.3 g ± 0.09) (Figure 2D).

### 3.2. Evaluation of ToLCNDV Symptomatology

At T2, all of the plants in the infected theses had developed the second true leaf, but no ToLCNDV symptoms were still evident. The symptoms appeared 10 days after inoculation in both IU and IT plants. As shown in Figure 4, at the end of the test (T4), the IT plants showed mosaic and slight curling of young and mature leaves (score = 2). The same symptoms were slightly more intense on IU plants, which also appeared more underdeveloped (score = 2). No recovery was observed in both theses.

### 3.3. Effect of Biostimulants on Viral Titer

Viral titer quantification was performed on infected leaf samples collected from the untreated and treated plants 24 h after the second (T2), third (T3) and fourth (T4) Fullcrhum application, corresponding to the third, fourth and sixth week after inoculation. In all inoculated plants, ToLCNDV infection was confirmed by qPCR. At T2, low numbers of ToLCNDV copies were detected in all of the plants in the infected theses (average copy numbers of 2 and 226 in IU and IT plants, respectively), indicating that the infection occurred systemically, despite the lack of symptoms. At T3, the viral titer increased to an average of 2.12 million copies in the untreated theses and 1.37 million copies in the treated ones. Then, the titer decreased after the last treatment in both IT (average of 1.06 million copies) and IU (average of 0.62 million copies) plants. At all experimental time points, the viral titer did not significantly differ between treated and untreated plants (Figure 5).

### 3.4. Effects of Biostimulants on Primary and Secondary Metabolism of Zucchini Squash Leaves

A total of 34 metabolites were identified and quantified from ^1^H-NMR spectra in hydroalcoholic and chloroform extracts. The ^1^H chemical shifts, multiplicity, and the ^13^C chemical shifts of the identified molecules are reported in Appendix A. The metabolic variations induced by biostimulants were assessed by fold-change analyses carried out on both healthy and infected samples collected at different treatment times. At T2, threonine, formic acid and total chlorophyll significantly increased after biostimulant treatment in healthy plants (Figure 6). Leucine, threonine, formic acid, glucose and choline increased in infected plants, whereas total chlorophyll decreased (Figure 6).

At T3, 13 metabolites significantly varied in healthy treated and untreated plants. These included leucine, threonine, alanine, γ-aminobutyric acid (GABA), glutamate, aspartate, asparagine, arginine, choline and total chlorophyll that increased and glutamine, tyrosine and phenylalanine that decreased after treatments. In the case of virus infection, beside threonine, glutamate, aspartate, asparagine, arginine and total chlorophyll, also valine, isoleucine, glutamine, tyrosine, phenylalanine, malic acid, chlorogenic acid, neo-chlorogenic acid, glucose, xylose, sucrose, raffinose, choline, uracil and trigonelline increased after treatments, while only formic acid showed a reduction (Figure 7).

At T4, threonine, alanine, glutamine, arginine, glucose, choline and trigonelline significantly increased in healthy treated plants, while leucine decreased (Figure 8). In the infected plants, most of the metabolites that positively changed at T3 were confirmed to significantly increase also after the T4 treatment, along with leucine, GABA, trigonelline, and stearic acid. Only linoleic acid decreased in infected treated plants at T4 (Figure 8).

## 4. Discussion

Among the plant pathogens, viruses are known to be responsible for great agronomic losses worldwide and always represent a major challenge to successful crop production. Thus far, therapeutic substances against plant viruses are not available or legally applicable, and their control is mainly based on preventive measures such as the insect-vector control and the use of genetically resistant cultivars [29]. Nevertheless, tolerant/resistant varieties or rootstocks are not always available, while an effective control of vectors often has a relevant environmental and economic impact. The control measures against ToLCNDV are very limited as well, and they mainly rely on the elimination of infected plants and whitefly control, whereas a few resistant or tolerant commercial zucchini cultivars became available only recently. Since its introduction, the virus rapidly spread throughout the Mediterranean area, and alternative measures have been constantly investigated to limit its circulation [20]. Good candidates to complement traditional agrochemicals are biostimulants, substances obtained from various economically and environmentally viable sources (e.g., seaweed extracts, humic and fulvic acids, amino acids, protein hydrolysates, chitin and chitosan derivatives, and microbes) whose market is constantly increasing worldwide [12,30].

Biostimulants generally induce positive effects on plant growth and productivity and are known to act as elicitors of the immune defenses or priming response making the plant more responsive to a wide range of stresses. However, little is known about their direct antiparasitic activity, and very limited data on their potential effects on plant viruses are available to date. Only single natural compounds have been observed to have a direct antiviral action, such as glucosylceramides and trichothecenes against tobacco mosaic virus (TMV) and pepper mottle virus (PepMoV), respectively [31,32]. Such activity has not been reported for complex mixtures so far, and possible applications of biostimulants in virus control are still at a theoretical stage [30,33,34]. In this work, the antiviral activity of commercially available biostimulants was tested against ToLCNDV in experimental conditions, and the possible beneficial effects on the fitness of infected zucchini plants were evaluated. Treatments with Fullcrhum Alert and BioVeg 500, combined with zeolite as a soil conditioner, did not affect virus replication and symptom expression. The systemic circulation of ToLCNDV was observed in both treated and untreated plants, as shown by the virus detection and quantification in new, completely expanded leaves that developed after the inoculation. Moreover, the virus titer did not significantly differ in treated and untreated plants after each biostimulant application, making the temporal trend of virus concentration independent of treatments. Regarding symptoms, both treated and untreated plants exhibited leaf mosaic, rib swelling, and a slight leaf curling, starting from 10 days post inoculation. At the end of the experimental trials, a comparable score of symptoms severity was assigned to the two groups of plants.

Nevertheless, a general beneficial effect was observed on the vegetative parameters of the treated plants, as previously reported [35,36,37]. After treatments, both healthy and ToLCNDV-infected plants always showed the greatest number of leaves and fruits, which were also characterized by a higher ripeness degree. This aspect is crucial, since it is known that in early infection, ToLCNDV could induce severe effects on flower/fruit development, as confirmed by comparing IU to IT. Additionally, biostimulants increased the dry weight of the plants. Particularly, the infected treated and healthy untreated plants showed a similar biomass content. This beneficial effect was even more evident in the healthy treated plants that showed a significantly higher biomass content than the healthy untreated ones. The obtained results showed that Fullcrhum does not act directly against ToLCNDV but rather influences the general well-being of the plants. As already reported for other biostimulants [38,39], Fullcrhum likely alters plant physiological processes, which, therefore, maintain their growth and productivity even in the presence of the virus infection. This result confirms the current knowledge that biostimulants can contribute to increases in plant vigor, but they do not necessarily act as antimicrobials [2,3].

The experimental data on plant fitness are supported by metabolomics analyses, which provided information on the possible regulation of plant metabolic networks by biostimulants. Most of the significant metabolic changes occurred in infected plants at T3 and T4. After these last two applications, the content in carbohydrates increased in the infected leaves. The accumulation of carbohydrates might be related to the higher biomass content observed in treated plants, and it is consistent with previous reports of the positive effect of biostimulants on carbohydrate metabolism [40]. In particular, the increase in sucrose concentration could be due to the higher photosynthetic rate as revealed by the increase in total chlorophyll content.

Moreover, other metabolic pathways appear to be enhanced in virus-infected plants. Specifically, an accumulation of various free amino acids was observed, including branched chain amino acids (BCAAs), such as valine, leucine and isoleucine, as well as other amino acids (e.g., alanine, glycine and asparagine), as previously reported for other plant/virus pathosystems [41]. The amino acids most likely accumulate in the stressed plants because of protein degradation induced by virus infection rather than de novo amino acid biosynthesis. The accumulated amino acids could serve as precursors for several secondary metabolites involved in signaling (e.g., hormones), structure (e.g., lignin), defense (e.g., glucosinolates), and protection (e.g., pigments) as response to virus infection. This hypothesis is supported by the increase in chlorogenic and neo-chlorogenic acid and phenylalanine in treated infected plants [42].

In this study, a variation of formic acid content was also observed both in healthy and infected plants. Particularly, an increase was recorded at T2, followed by a decrease at T3. Nevertheless, in treated infected plants, this reduction was stronger than in healthy ones. Interestingly, this reduction matched with the pick of viral replication observed at T3, suggesting a possible involvement of this acid in plant defense response. Formic acid may have a role in the biosynthesis of numerous compounds, in energetic metabolism and in signal transduction pathways related to stress response [43].

Effects of biostimulant treatments were observed on the regulation of fatty acid metabolism. At T4, the increase in saturated stearic acid was associated with the decrease in unsaturated linoleic acid, with possible consequences for the stocks of extracellular barrier constituents (e.g., cutin and suberin), the biosynthesis of various bioactive molecules (e.g., jasmonates and nitroalkenes) and regulators of stress signaling [44]. At both T3 and T4, the infected plants were affected by the biostimulant treatment much more than at T2. Healthy treated plants showed a similar trend, but less marked and for a reduced number of metabolites. The defensive pathways represented by metabolic reprogramming of chlorogenic and neo-chlorogenic acid and unsaturated and saturated fatty acids were not stimulated in healthy treated plants.

A high level of trigonelline, a signal transmitter in the response to oxidative stress, was also found in infected plants after treatment but not in healthy ones. Trigonelline may function as signal transmitter in the response of plants to oxidative stress induced by virus infection [45]. Synthesis of GABA was also observed after treatments as revealed by the increase in this metabolite in healthy plants at T3. This increase was not recorded in infected plants at T3, corresponding to the peak of viral replication, providing evidence of possible involvement of GABA in metabolic pathways linked to the response of plants to virus infection [46]. As expected, metabolomics analysis in treated plants revealed an increase in betaine (Appendix A), which was supplied through Fullcrhum BioVeg 500. However, also the choline, a precursor of the glycine betaine, was shown to increase. Since choline is needed to synthesize the membrane phospholipid phosphatidylcholine, its increase further contributed to the osmoprotectant activity of the glycine betaine [47]. The overall metabolomic results showed that the Fullcrhum application enhanced those metabolic pathways that are commonly at the base of plant response to virus infection.

## 5. Conclusions

The vegetative parameters observed after biostimulant application provide evidence on the ability of the treated plants to survive in the presence of virus and to remain productive. These findings are of relevance in cases of virus diseases for which a direct control is not applicable, as for ToLCNDV. This supports the opportunity to incorporate the use of biostimulants into horticultural production systems. Moreover, due to their very low or absent side effects, the use of such compounds can ensure safe agricultural practices and green horticultural production. In this sense, a field-scale study is necessary to better clarify and confirm the effect of Fullcrhum on zucchini productivity observed in experimental conditions, and to ascertain possible market feedbacks. This study also indicates that Fullcrhum contains biologically active molecules that enhance various physiological responses to stresses induced by infection. Further molecular studies will be needed to investigate the possible genetic pathways that are responsible for these defense responses and to identify differentially expressed genes.

## Figures and Tables

**Figure 1 viruses-14-00607-f001:**
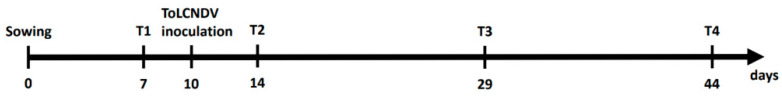
Timing of biostimulant treatments and tomato leaf curl New Delhi virus inoculations on zucchini plants used in the experimental design.

**Figure 2 viruses-14-00607-f002:**
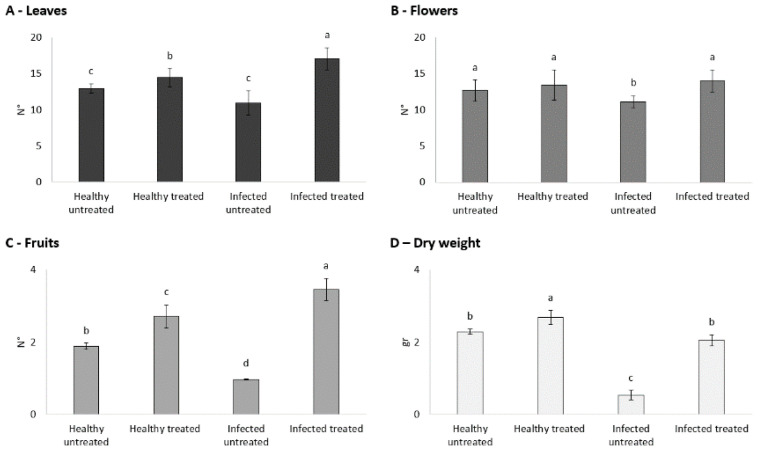
Number of leaves (**A**), flowers (**B**), fruits (**C**), and dry weight (**D**) of healthy and tomato leaf curl New Delhi virus-infected zucchini squash plants, treated and untreated. Values are expressed as mean ± standard error for 15 biological replicates ×3 independent repeats. Different letters indicate statistically significant differences at *p* < 0.05. Significantly different groups were identified using Fisher’s least significant difference (LSD) *post hoc* test.

**Figure 3 viruses-14-00607-f003:**
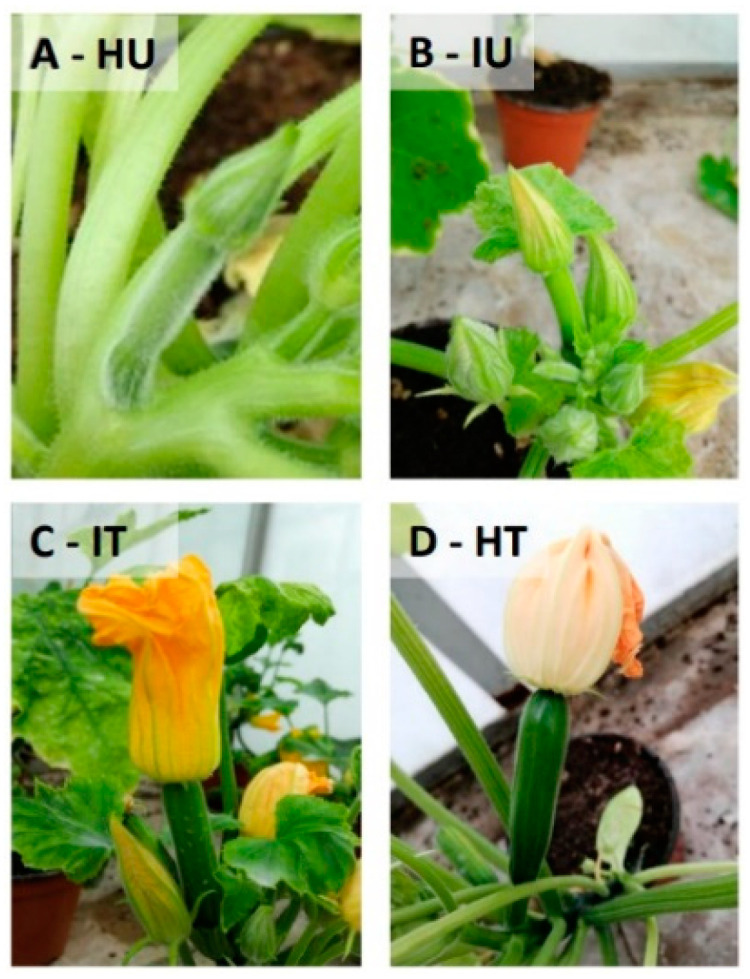
Development and ripeness level of fruits produced at the end of the experiment by: (**A**) healthy untreated plants; (**B**) ToLCNDV-infected untreated plants; (**C**) ToLCNDV-treated plants; (**D**) healthy treated plants.

**Figure 4 viruses-14-00607-f004:**
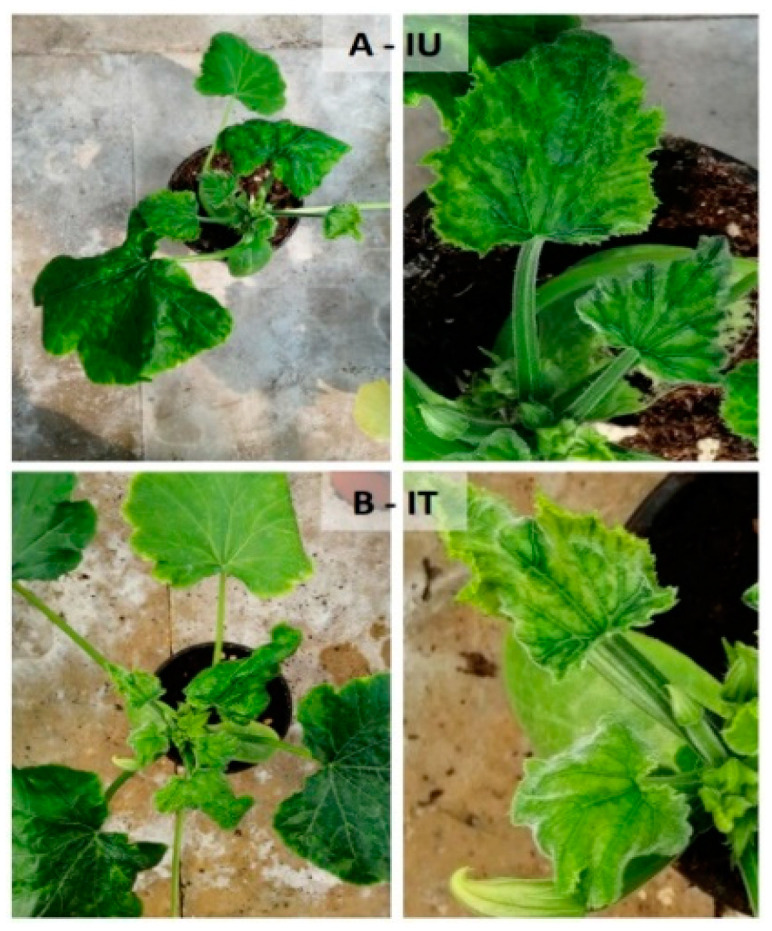
Symptoms of tomato leaf curl New Delhi virus infection on untreated (IU) zucchini plants (**A**) and Fullcrhum-treated infected (IT) plants (**B**) at T4. Details of symptoms are shown on the right panels.

**Figure 5 viruses-14-00607-f005:**
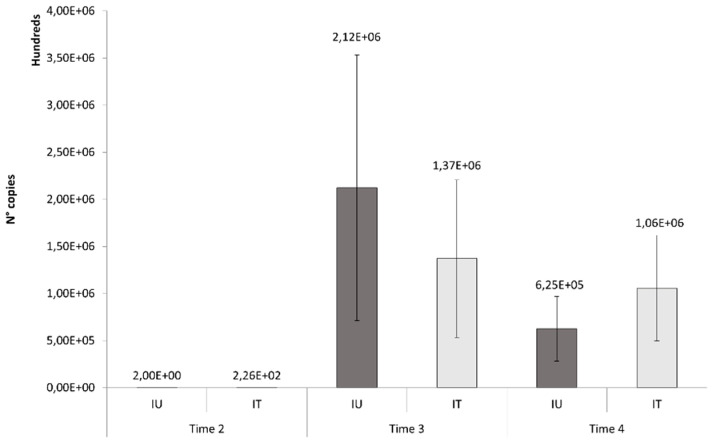
Tomato leaf curl New Delhi virus titer quantification in treated (IT) and untreated (UT) zucchini squash plants. Differences in the number of virus copies were not significant by the Student’s *t*-test (*p* > 0.05).

**Figure 6 viruses-14-00607-f006:**
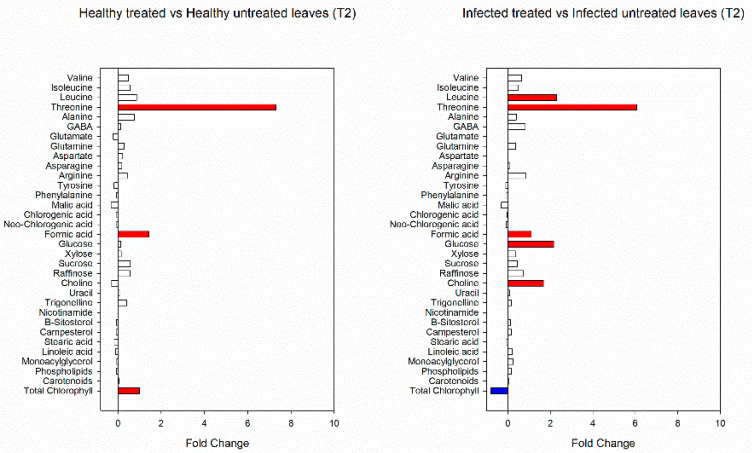
Fold changes of identified and quantified metabolites in healthy and infected plants treated with biostimulants at T2. Fold-change cut-off = 1.5. Red color indicates a fold-change higer than 1.5; blue color indicates a fold change less than 1.5.

**Figure 7 viruses-14-00607-f007:**
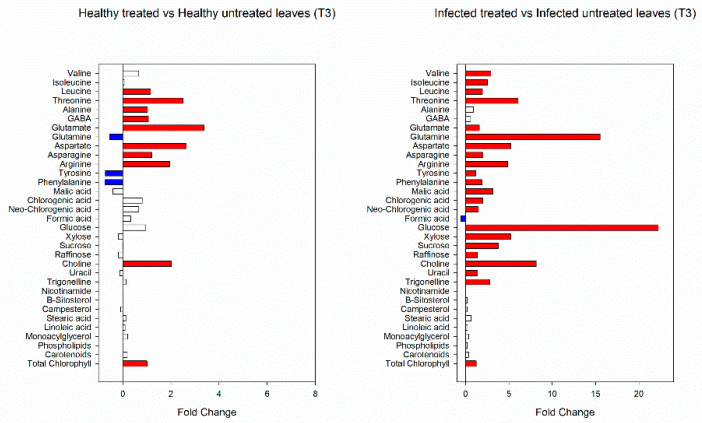
Fold changes of identified and quantified metabolites in healthy and infected plants treated with biostimulants at T3. Fold-change cut-off = 1.5.Red color indicates a fold-change higer than 1.5; blue color indicates a fold change less than 1.5.

**Figure 8 viruses-14-00607-f008:**
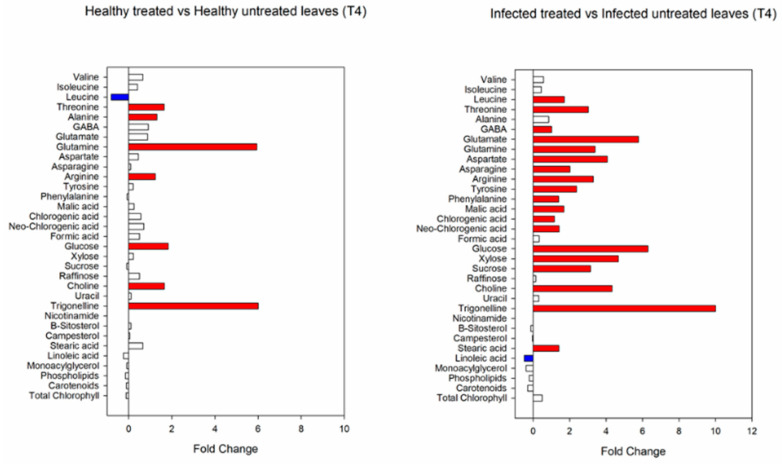
Fold changes of identified and quantified metabolites in healthy and infected plants treated with biostimulants at T4. Fold-change cut-off = 1.5.Red color indicates a fold-change higer than 1.5; blue color indicates a fold change less than 1.5.

**Table 1 viruses-14-00607-t001:** Classes of symptoms associated with tomato leaf curl New Delhi virus infection, descriptions of the degrees of severity, and assigned scores.

Symptoms	Description	Score
Absent	No symptoms	0
Slight	Slight leaf mosaic	1
Medium	Leaf mosaic and curling	2
Severe	Rib swelling, accentuated leaf mosaic and curling, reduced development of the plant	3

## Data Availability

Not applicable.

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
