# Peer review of "Effects of Organic Biostimulants Added with Zeolite on Zucchini Squash Plants Infected by Tomato Leaf Curl New Delhi Virus"

_viruses, 2022, doi:10.3390/v14030607_

Round 1
Reviewer 1 Report
I have a few questions on areas that I felt needed further clarification.
- I think the reader needs to know what "plant fitness" means in the context of this study. This was mentioned multiple times with plant growth and development, indicating it has a different meaning to these two. If possible, please define this term.
- Were there any tests conducted to determine that ToLCNDV was the only (viral) pathogen present in the infectedplants? Begomoviruses are often found in mixed infections.
- Was DNA isolated from a single leaf? Is this representative enough of virus distribution in the plant?
- Was the "second true leaf" the source of DNA for all sampling points, or just the first one (T2)?
- How many leaves were used for the metabolomics part of the study? Any specific leaf position?
Please also see the additional comments I left in the manuscript.

Author Response
1. I think the reader needs to know what "plant fitness" means in the context of this study. This was mentioned multiple times with plant growth and development, indicating it has a different meaning to these two. If possible, please define this term.
Response: The term “fitness” has been defined in the last paragraph of the Introduction section.
2. Were there any tests conducted to determine that ToLCNDV was the only (viral) pathogen present in the infected plants? Begomoviruses are often found in mixed infections.
Response: We tested the source of ToLCNDV-ES inoculum for those potyviruses that are commonly found in cucurbits in our area, to exclude mixed infection. The following sentence has been added to Materials and Methods section: “The zucchini plant was tested for the most common cucurbit potyviruses in the Mediterranean areas (Zucchini yellow mosaic virus, Watermelon mosaic virus, Cucumber mosaic virus) to exclude any mixed infections.” We did not test the sample for other begomoviruses since only two species are known to be present in Italy (Tomato yellow leaf curl virus and Tomato Sardinian yellow leaf curl virus) and they infect only Solanaceous plants. Once we excluded any mixed infection in the source of inoculum, we did not test all the plants used in the experiments since these were mechanically inoculated and grown in a screen-house in controlled conditions (absence of insects and other begomovirus-infected plants).
3. Was DNA isolated from a single leaf? Is this representative enough of virus distribution in the plant?
Response: The Material and Methods section has been modified specifying that DNA was extracted from a pool of three leaves.
4. Was the "second true leaf" the source of DNA for all sampling points, or just the first one (T2)?
Response: The Material and Methods section has been modified specifying that DNA was extracted from the last true leaf.
5. How many leaves were used for the metabolomics part of the study? Any specific leaf position?
Response: A total of three leaves of the same size (total weight: 0.5 grams) were used for Bligh-dyer extraction in metabolomics analysis.
All the other modifications suggested by the reviewer in the main text have been accepted.
Reviewer 2 Report
The manuscript reports the effect of biostimulant agents on growth and development of zucchini plants that were infected with the begomovirus Tomato leaf curl New Dehli virus (ToLCNDV). Specific vegetative parameters, as well as symptomatology and virus concentration were recorded in order to investigate the performance of different organic compounds on infected plants. The authors have also investigated the presence of several metabolites which were accumulated after the treatments, using spectroscopic methodology.
Although methodology and results are quite well and detailed presented, the Introduction and the Discussion sections could be further improved.
A few suggestions for manuscript improvement can be found below:
TITLE: could be shortened
Effects of organic biostimulants on zucchini squash plants infected by Tomato leaf curl New Delhi virus (ToLCNDV)
ABSTRACT
LINE 16. Fullchrum: This word is can be found with different spelling in the article. Is it Fullcrhum or Fullchrum?
INTRODUCTION
General comments: This section is too long and could be reduced. If Methodology aspects are mentioned in the Material and Methods section, they could be removed from Introduction.
Below are some comments and suggestions for authors.
LINE 48. ….. because they contain bioactive compounds….
LINE 50. Fullcrhum or Fullchrum ?
LINE 54. These biostimulants are often used in combination with zeolite, a soil conditioner that shows a positive effect in……
LINES 58-59. few information is available on the effect of systemic infection by plant pathogens.
LINES 63-87. Use new paragraph and remove unnecessary information
Cucurbits such as melon (Cucumis melo L.), watermelon (Citrullus lanatus Thunb.), cucumber (Cucumis sativus L.), pumpkin species (Cucurbita moschata D. and C. maxima D.) and zucchini (Cucurbita pepo L.) are major agricultural crops in the Mediterranean basin. Several virus diseases have been reported to infect cucurbits, having a negative impact on greenhouse and open field production. The recent emergence of Tomato leaf curl New Delhi virus (ToLCNDV) (genus Begomovirus, family Geminiviridae) in the Mediterranean region has caused major problems on crop production and quality. The virus is transmitted by the whitefly Bemisia tabaci Gennadius (Hemiptera: Aleurodidae) in a persistent manner, and until now it has been reported in Italy, Spain, France, Portugal, Greece, Tunisia and Algeria [18, 22-25]. Infected plants show symptoms of yellow mosaic, leaf curling, vein swelling and plant stunting. On cucurbits fruit skin may present a rough appearance with longitudinal cracking [17,18].
LINE 88. Use new paragraph and state the aim of this work very briefly.
This study aims to evaluate …..of disease prevention and protection.
LINES 90-100 include Methodology and could be removed.
MATERIALS AND METHODS
General comment: Perhaps the authors could describe methodology at parts 2.4 and 2.5 more briefly.
LINE 108: ….during the summer of 2019…
LINE 115: Fullcrhum or Fullchrum ?
LINES 146 - 147: virus trite ?
DISCUSSION
The authors are encouraged to revise this section, include paragraphs, and perhaps provide a better discussion of the results, in accordance with the existing literature. A few grammatical errors that appear in the text could also be corrected.
Author Response
Although methodology and results are quite well and detailed presented, the Introduction and the Discussion sections could be further improved.
A few suggestions for manuscript improvement can be found below:
TITLE: could be shortened
1. Effects of organic biostimulants on zucchini squash plants infected by Tomato leaf curl New Delhi virus(ToLCNDV)
Response: Title as been modified as follow: “Effects of organic biostimulants added with zeolite on zucchini squash plants infected by Tomato leaf curl New Delhi virus”
ABSTRACT
2. LINE 16. Fullchrum: This word is can be found with different spelling in the article. Is it Fullcrhum or Fullchrum?
Response: The correct name is Fullcrhum. The authors checked and modified this word in all the manuscript.
INTRODUCTION
General comments: This section is too long and could be reduced. If Methodology aspects are mentioned in the Material and Methods section, they could be removed from Introduction.
The Introduction has been shortened as suggested and the last part of this section was moved to Materials and Methods.
Below are some comments and suggestions for authors.
3. LINE 48. ….. because they contain bioactive compounds….
Response: Text modified as suggested
4. LINE 50. Fullcrhum or Fullchrum?
Response: This word was corrected with the proper spelling
5. LINE 54. These biostimulants are often used in combination with zeolite, a soil conditioner that shows a positive effect in……
Response: Text modified as suggested
6. LINES 58-59. few information is available on the effect of systemic infection by plant pathogens.
Response: Text modified as suggested
7. LINES 63-87. Use new paragraph and remove unnecessary information
Cucurbits such as melon (Cucumis melo L.), watermelon (Citrullus lanatus Thunb.), cucumber (Cucumis sativus L.), pumpkin species (Cucurbita moschata D. and C. maxima D.) and zucchini (Cucurbita pepo L.) are major agricultural crops in the Mediterranean basin. Several virus diseases have been reported to infect cucurbits, having a negative impact on greenhouse and open field production. The recent emergence of Tomato leaf curl New Delhi virus (ToLCNDV) (genus Begomovirus, family Geminiviridae) in the Mediterranean region has caused major problems on crop production and quality. The virus is transmitted by the whitefly Bemisia tabaci Gennadius (Hemiptera: Aleurodidae) in a persistent manner, and until now it has been reported in Italy, Spain, France, Portugal, Greece, Tunisia and Algeria [18, 22-25]. Infected plants show symptoms of yellow mosaic, leaf curling, vein swelling and plant stunting. On cucurbits fruit skin may present a rough appearance with longitudinal cracking [17,18].
Response: This paragraph has been modified as suggested.
8. LINE 88. Use new paragraph and state the aim of this work very briefly.
This study aims to evaluate …..of disease prevention and protection.
Response: The paragraph has been modified as suggested.
9. LINES 90-100 include Methodology and could be removed.
Response: Text from lines 90 to 100 has been removed.
MATERIALS AND METHODS
10. General comment: Perhaps the authors could describe methodology at parts 2.4 and 2.5 more briefly.
Response: Both paragraph have been reframe in a more brief form.
11. LINE 108: ….during the summer of 2019…
Response: The sentence has been modified.
12. LINE 115: Fullcrhum or Fullchrum ?
Response: Modified
13. LINES 146 - 147: virus trite?
Response: Modified with “titre”
DISCUSSION
14. The authors are encouraged to revise this section, include paragraphs, and perhaps provide a better discussion of the results, in accordance with the existing literature. A few grammatical errors that appear in the text could also be corrected.
Response: The authors have checked the language through the manuscript and corrected the grammar errors. Discussion section has been revised including paragraphs, reducing some comments about the metabolomic results, and adding some references to the existing literature. Concerning this point, the authors point out that data on the use of biostimulants against plant viruses are poorly available.
Reviewer 3 Report
- I will suggest authors to modify the title of manuscript. Include the specific organic biostimulant used also the inorganic substance used in the study.
- I will suggest authors to check the language of manuscript thoroughly or take the help of any native English speaker.
- I will suggest authors to modify the abstract part using short clear sentences. Include the major findings with numerical value instead of methodology. Please see the examples of research papers published in the Journal.
- Include the size of pots used in the study. Include the kind of matrix used to support the plant.
- I am not satisfied with construction of manuscript. I will suggest authors to read the published research papers and reconstruct the manuscript.

Author Response
Comments and Suggestions for Authors
1. I will suggest authors to modify the title of manuscript. Include the specific organic biostimulant used also the inorganic substance used in the study.
Response: Title has been modified as follow, as suggested by reviewer 2: “Effects of organic biostimulants added with zeolite on zucchini squash plants infected by Tomato leaf curl New Delhi virus”. The authors did not include the specific names of biostimulants because, in our opinion, it is not correct to insert the name of the commercial product in the title of a scientific article.
2. I will suggest authors to check the language of manuscript thoroughly or take the help of any native English speaker.
Response: The authors have checked the language through the manuscript and corrected the grammar errors.
3. I will suggest authors to modify the abstract part using short clear sentences. Include the major findings with numerical value instead of methodology. Please see the examples of research papers published in the Journal.
Response: The abstract has been fully revised.
4. Include the size of pots used in the study. Include the kind of matrix used to support the plant.
Response: Size of pots and the name of the used matrix has been added.
5. I am not satisfied with construction of manuscript. I will suggest authors to read the published research papers and reconstruct the manuscript.
Response: The authors reviewed the manuscript and tried to improve the lacking sections in Introduction and Discussion (also following the reviewer 2 suggestion). More citations have been added to better align the study with existing literature. The authors wish that this new version can satisfy the reviewer. If the reviewer wants to specify any other suggestion on manuscript construction the authors are willing to accept them.
Abstract
6. Line 15-16: Re-frame the sentence
Response: The abstract was re-framed following the suggestion of the reviewers.
Introduction:
7. Line 90-99: This should be the part of methodology. Introductory last paragraph should include why the study done after seeing the literature.
Response: In particular, the experimental trials were carried out using a combination of Fullcrhum Alert, Fullchrum BioVeg 500 and Clinogold zeolite (BioHelp Your Planet srl, Viterbo, Italy). The treatments were carried out on both healthy and ToLCNDV-infected plants and possible effects on both plant fitness and virus titre were assessed. Since the mode of action of biostimulants is still largely unknown, an 1H nuclear magnetic resonance (NMR) spectroscopy as an untargeted approach was used to elucidate the metabolic changes occurring in leaves of zucchini squash plants associated with biostimulants application and the infection process mediated by ToLCNDV. To our knowledge, this is the first report of metabolic profiling by 1H NMR spectroscopy in zucchini plants infected with ToLCNDV and/or treated with biostimulants.
Response: The authors deleted these sentences and reduced Introduction section.
Methods:
8. Line 154-155: The qPCR standard curve was constructed on a DNA-A fragment of about 1500 bp of the ToLCNDV-ES isolate used for the experimental inoculations. Mention qPCR conditions.
Response: A reference to the qPCR conditions has been added in the text.
9. Figure 2: Number of leaves (A), flowers (B), fruits (C), and dry weight (D) of healthy and tomato leaf curl New Delhi virus-infected zucchini squash plants, treated and untreated. Values are expressed as mean ± standard error for 15 biological replicates x 3 independent repeats. Different letters indicate statistically significant differences at p<0.05. Please check the letters representing significance of the error bars (in one bar diagram letter a is presenting highest however in other its least). Include the name of test performed in the figure legends.
Response: Figure 2 has been modified as suggested by the reviewer. The name of test performed has been added to the legend.
10. Figure 5: Tomato leaf curl New Delhi virus titre quantification in treated (IT) and untreated (UT) zucchini squash plants. Differences in the number of virus copies were checked not significant by the Student's t-test. include level of significance.
Response: p-value (>0.05) was added to the legend.
All the other modifications suggested by the reviewer in the main text have been accepted.